# Expert-level Leaf Cell Layout Generation via Preference-Optimized LLM

Yaohui Han[1]    Rongliang Fu[*1]    Yanming Liu[2]    Shuo Ren[1]    Shuai Dong[2]    Yunpeng Wang[2]
Tinghuan Chen[3]    Bei Yu[1]    Tsung-Yi Ho[1]

## Abstract

In the field of integrated circuits, leaf cells are the basic units, serving as the fundamental building blocks (e.g., standard cells) that are widely reused in various VLSI designs, forming the basis for more complex circuits. Therefore, the design quality of leaf cell layouts significantly impacts the PPA (Power, Performance, and Area) of the final VLSI designs. To automatically design leaf cell layouts that are close to expert designs, we propose GenLeaf. GenLeaf first utilizes a supervised, performance-aware embedding model to represent layouts and automatically calculate their similarity scores. Since there are expert-designed layouts but no corresponding scripts, we implement Bayesian optimization to generate a layout-script dataset for LLM training. With subsequent supervised fine-tuning and further preference optimization, GenLeaf can generate leaf cell layouts through scripts whose performance closely resembles that designed by human engineers. Experimental results demonstrate that GenLeaf outperforms expert-designed golden layouts across key performance metrics.

## 1. Introduction

With the growth of microelectronics, the structures of a single chip are becoming increasingly complex. Typically, they are composed of the smallest artificially designed units, called leaf cells (Bamji & Varadarajan, 2012). Ranging from standard cells to memory bitcells (Lee et al., 2023; Kim et al., 2025a) and analog components, each leaf cell is deeply reused across VLSI designs. Therefore, the design quality of these leaf cells, especially physical design,

[1]Department of Computer Science and Engineering, The Chinese University of Hong Kong [2]Huawei Technologies Co., Ltd [3]School of Science and Engineering, The Chinese University of Hong Kong, Shenzhen. *Correspondence to: Rongliang Fu <rlfu@cse.cuhk.edu.hk>.

*Proceedings of the 43rd International Conference on Machine Learning*, Seoul, South Korea. PMLR 306, 2026. Copyright 2026 by the author(s).

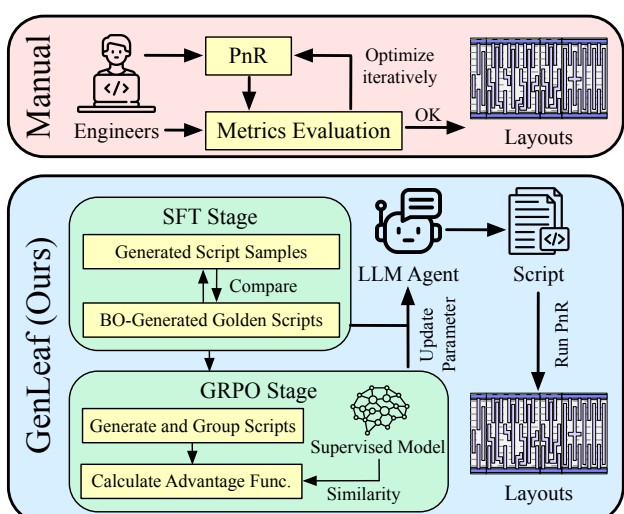

*Figure 1.* Comparison between the proposed GenLeaf and the traditional manual design method for leaf cell layout design.

greatly influences the Power, Performance, and Area (PPA) metrics of the entire chip design. In the physical design of leaf cells, track usage, wirelength, and via count serve as critical proxies for PPA, as they directly determine parasitic parameters and silicon area (Zhang et al., 2025).

However, designing high-quality leaf cell layouts is quite challenging. In custom Integrated Circuit (IC) design, numerous fine-grained metrics require simultaneous optimization (Kahng et al., 2011). Since these metrics are mutually dependent, even slight design alterations can significantly impact the overall circuit performance. Therefore, such custom layouts are still primarily designed manually by experts to achieve optimal results. For instance, in typical DRAM peripheral circuits, custom digital components utilize over ten times the number of transistors compared to analog circuits (Kim et al., 2025b). These components rely on deeply reused leaf cells (usually containing dozens of logic gates). Despite their moderate scale, the design space is vast, necessitating extensive iterations to meet ideal specifications. Currently, this process is fully manual, time-consuming, and requires high expertise, as shown in Figure 1. Furthermore, while design knowledge is inherently reusable, existing software struggles to transfer this tacit knowledge across different designers or adjacent projects. From an

automation perspective, the challenges lie in the complex design space of Placement and Routing (PnR) and the lack of effective early evaluation (Xie et al., 2022; Han et al., 2026a). Specifically, each module in the leaf cell can be rotated, flipped, and placed freely. Meanwhile, there are usually multiple metal layers for leaf cell routing, so the solution space of leaf cell layout is extremely large. In addition, the evaluation of leaf cell placement depends on indirect metrics (e.g., predicted wirelength and density), which cannot demonstrate the overall performance of leaf cell layouts after routing.

Various algorithms have been developed to solve the challenges of leaf cell layout design. Hougardy et al. introduced BonnCell (Hougardy et al., 2013) to automate leaf cell layout generation while maintaining design rules. Duran et al. proposed a placement algorithm based on Boolean Satisfiability (SAT) formulations (Duran & Roa, 2021) with routing optimization. Chu et al. utilized Linear Programming (LP) and Mixed Integer Linear Programming (MILP) (Sham et al., 2006) to solve the cell flipping problem.

Despite these advancements, traditional methods and commercial tools (e.g., Pulsic Unity (Pulsic, 2023)) often rely on rigid heuristics. While claiming to accelerate iterations by considering routing during placement, they struggle to capture the implicit design patterns of experts, yielding results that differ significantly from human-crafted quality. These tools cannot learn from historical data or typically fail to handle complex constraints flexibly. Consequently, current industry workflows (e.g., Virtuoso (Cadence, 2026), Aether (Empyrean Technology, 2025)) still heavily depend on layout engineers to manually translate designer intent into physical implementation. On the other hand, the rapid development of Artificial Intelligence (AI) provides new possible solutions for Electronic Design Automation (EDA) (Zhang et al., 2025; Zhu et al., 2022; Yang et al., 2022).

To address these complexity and reusability challenges in custom IC and DRAM design, Large Language Models (LLMs) offer unique advantages and have been initially applied in the EDA problems (Liu et al., 2025; Zhao et al., 2025a; Wu et al., 2025; Han et al., 2026b). Their massive parameter scale enables them to model complex interdependencies and fine-grained metrics better than smaller models. Through Supervised Fine-Tuning (SFT) and Group Relative Policy Optimization (GRPO) on expert-designed layouts, LLMs can absorb tacit design knowledge and continuously improve generation quality. Moreover, their ability to process natural language allows for flexible design adjustments based on specific requirements, effectively bridging the gap between schematic intent and physical realization.

To the best of our knowledge, GenLeaf is the first LLM-based framework dedicated to generating expert-level leaf cell layouts. Overall, our main contributions are as follows:

- We propose GenLeaf, a novel framework that leverages LLMs to automate the design of leaf cell layouts. By treating layout generation as a script generation task, GenLeaf bridges the gap between high-level design intent and physical implementation.

- We design a performance-aware representation learning method that combines graph neural networks and CNNs to capture both topological and routing features, enabling automated early evaluation of layout quality.

- We develop a Bayesian Optimization (BO) based data preparation pipeline to reverse-engineer expert layouts into scripts, overcoming the data scarcity of high-quality layout-script pairs.

- We propose a two-stage specialization strategy involving SFT and GRPO, allowing the LLM to learn tacit design knowledge and align with human expert preferences, thereby enhancing the overall performance.

- Extensive experiments show that GenLeaf outperforms both traditional methods and SOTA LLMs, producing layouts that rival or exceed human expert quality in terms of track usage, wirelength, and via count.

## 2. Problem Formulation

**Inputs** A netlist $\mathcal{N}(C, E)$ containing a set of component cells $C$ and their connectivity $E$, and a set of design rules covering physical constraints.

**Output** A geometric layout $L$ (generated via script $s$) containing the physical placement and orientation of all cells, as well as the routing paths connecting them.

**Constraints** The layout must strictly satisfy two sets of design rules:
1) **Placement Rules**: All cells must be placed within the layout boundary without overlapping. The orientation of each cell is restricted to $\mathcal{O}_{set} = \{R0, MY\}$.
2) **Routing Rules**: All terminals of the same net must be electrically connected (LVS correct). The geometry of wires and vias must meet physical design rules (e.g., minimum width, spacing) to be DRC-clean.

**Goal** Our objective is to train a generative model $\pi_\theta$ that takes a design query $q$ and outputs a script $s$, such that the resulting layout $L_s$ satisfies all constraints and minimizes the cost function $\mathcal{C}(L)$:

$$\mathcal{C}(L) = \alpha \cdot \text{Track} + \beta \cdot \text{WL} + \gamma \cdot \text{Via}, \quad (1)$$

where WL is wirelength, Via is the via count, and Track is the number of used routing tracks. Besides, $\alpha$, $\beta$, and $\gamma$ are the hyperparameters to balance the impact of the three physical metrics. Among these

*Table 1.* Definition of vertex feature $\mathcal{F}$ in the proposed PHG.

| Feature | Type | Definition |
|---------|------|-----------|
| $\mathcal{F}_c$ | Cell | Height & Width
One-hot encoding
Number of pins |
| | Geometric | Pin coordinates
Rotation $\mathcal{R}$ |
| $\mathcal{F}_n$ | Connection | Net degree |
| | Span | Net span in Y axis
Net span in X axis |

metrics, the number of tracks has the greatest impact on the final VLSI design quality. Fewer tracks allow for more routing resources in subsequent designs, thus significantly improving overall performance.

## 3. Algorithm

### 3.1. Leaf Cell Layout Representation

#### 3.1.1. PLACEMENT TOPOLOGY EXTRACTION

To model spatial and connection relationships between cells within the leaf cell layouts, we propose the placement heterogeneous graph (PHG). As illustrated in Figure 2, this graph explicitly combines both physical adjacency among cells and topological connection through nets. The vertex set $\mathcal{V}$ comprises cell vertices $\mathcal{V}_c$ and net vertices $\mathcal{V}_n$. As for the edge set $\mathcal{E}$, there are two types of connection: 1) Cell-net connection: an edge $(v_c, v_n)$ exists if cell $c$ hosts a pin belonging to net $n$. 2) Cell-cell spatial adjacency: an edge $(v_{c_1}, v_{c_2})$ connects cell vertices $v_{c_1}$ and $v_{c_2}$ if their physical placements are adjacent, including sharing a boundary or within a Manhattan distance threshold $\delta$. For each vertex in PHG, the features of cells and nets are represented as $\mathcal{F}_c$ and $\mathcal{F}_n$, respectively, as shown in the Table 1. In this way, the features of the proposed PHG can better support downstream physical design tasks.

#### 3.1.2. ROUTING PATTERN EXTRACTION

Routing constitutes a critical component in integrated circuit layout design. In the existing layout representation frameworks for VLSI, routing information is often either highly abstracted or omitted altogether, primarily because of the complexity of full routing extraction. Therefore, these representation methods cannot provide a satisfying extraction and representation of routing features. However, in leaf cell layouts, where design customization is typically more stringent and structural complexity is relatively lower, it becomes vital to accurately capture and represent detailed routing features to ensure precise modeling and reliable performance of downstream tasks.

**Algorithm 1** GraphSAGE-based PHG Processing

**Require:** PHG $\mathcal{G}(\mathcal{V}, \mathcal{E})$; Vertex features $\mathcal{F}$; Depth $K$; Neighborhood sample sizes $\{S_k\}_{k=1}^K$; Learnable weights $\{\mathbf{W}_{\text{self}}^k, \mathbf{W}_{\mathcal{N}}^k\}_{k=1}^K$
1: $\mathbf{h}_v^0 \leftarrow \text{Linear}(\mathcal{F}_v), \forall v \in \mathcal{V}$     // Feature projection
2: **for** $k \in (1, K)$ **do**
3:    **for** $v \in \mathcal{V}$ **do**       // SAGEConv layer
4:      $\mathcal{N}_s(v) \leftarrow \text{SAMPLE}(\mathcal{N}(v), S_k)$
5:      $\mathbf{h}_{\mathcal{N}(v)}^k \leftarrow \frac{1}{|\mathcal{N}_s(v)|} \sum_{u \in \mathcal{N}_s(v)} \mathbf{h}_u^{k-1}$
6:      $\mathbf{h}_v^k \leftarrow \mathbf{W}_{\text{self}}^k \cdot \mathbf{h}_v^{k-1} + \mathbf{W}_{\mathcal{N}}^k \cdot \mathbf{h}_{\mathcal{N}(v)}^k$
7:    **end for**
8:    $\mathbf{h}^k \leftarrow \text{Dropout}(\sigma(\text{BatchNorm}(\mathbf{h}^k)))$
9:    **if** $k > 1$ **then**
10:     $\mathbf{h}^k \leftarrow \mathbf{h}^k + \mathbf{h}^{k-1}$     // Residual connection
11:    **end if**
12: **end for**
13: $\mathbf{P} \leftarrow \text{MLP}(\text{CONCAT}(\text{MeanPool}(\mathbf{h}^K), \text{MaxPool}(\mathbf{h}^K)))$
14: **return P**

Because the routing complexity is relatively simple, we can represent it from a more refined perspective. In GenLeaf, the routing pattern of leaf cell layouts is treated as image inputs. Specifically, the whole routing image can be divided into three categories: net, track, and blank area. Net region refers to the area occupied by wire segments and vias. Notably, locations covered by different nets correspond to different values in the routing image. Track regions are the areas that can be used for routing. And the remaining areas all belong to the blank regions. As for the multi-layer circumstances, each pixel in the routing image corresponds to a vector, and the elements in this vector correspond to the routing patterns of different layers. This extraction method of routing can describe the layouts in more detail, thereby significantly improving the quality of downstream tasks.

#### 3.1.3. SUPERVISED FEATURE PROCESSING AND FUSION

**GraphSAGE-based PHG Processing.** After extracting the feature of leaf cell layouts, GenLeaf then performs feature processing and fusion to generate the final layout embedding. Since the topology information of placement has been well extracted into the proposed PHG format, we implement a GraphSAGE-based network (Hamilton et al., 2017) to process the PHG, focusing on capturing the relative positional relationships between different cells. As shown in Algorithm 1, first, each vertex feature is processed by a linear layer (line 1). Then, all features are processed by SAGEConv: each SAGEConv first performs neighborhood sampling and then aggregates the sampled features using specified weights (lines 5-7). And then we apply dropout and residual connection to enhance the performance of the model (lines 9-12). After $K$ iterations, we perform mean-pooling and max-pooling on the processed features and get

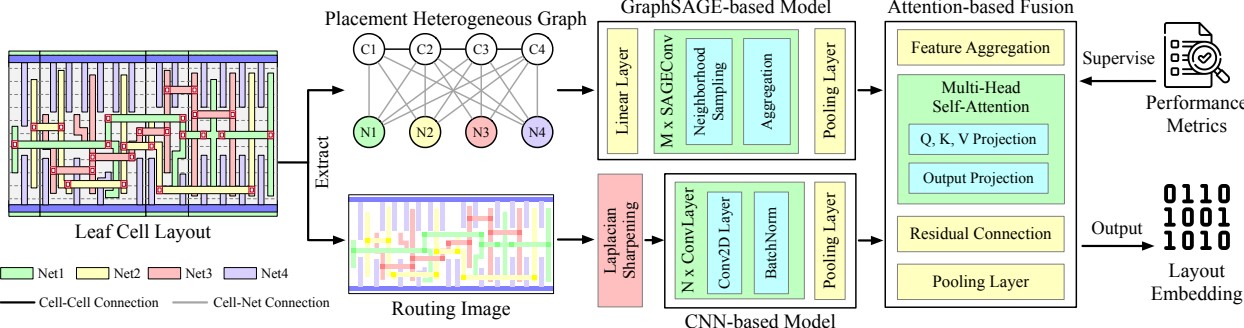

*Figure 2.* The supervised leaf cell layout representation.

the final placement features for subsequent representation (lines 14-15).

**CNN-based Feature Extraction for Routing.** As for routing, GenLeaf utilizes a Laplacian sharpening scheme to highlight the boundaries of different regions and a CNN to embed the processed routing image. Clearer boundaries between regions can improve the effectiveness of routing features, thereby improving the overall performance of GenLeaf. The values in the routing images do not change continuously, so we use the discrete Laplacian sharpening:

$$g(x, y) = f(x, y) - c \cdot \nabla^2 f(x, y), \quad (2)$$

where $g(x, y)$ and $f(x, y)$ represent the sharpened and original images, respectively. $c$ is a hyperparameter that determines the strength of sharpening. $\nabla^2 f(x, y)$ is the Laplacian operator, defined in Equation (3).

$$\nabla^2 f(x, y) = f(x + 1, y) + f(x - 1, y) + f(x, y + 1) + \\ f(x, y - 1) - 4 \cdot f(x, y) \quad (3)$$

**Performance-aware Supervised Representation.** However, evaluating layouts solely based on the topology similarity is insufficient and, even when combined with other performance metrics, may lead to optimization efforts that deviate from the intended direction. Therefore, we propose performance-aware supervised representation learning. As shown in Figure 2, since a vital objective of GenLeaf is to design leaf cell layouts with shorter wirelength, fewer vias and used tracks, we select the three metrics to form the loss function. Therefore, layouts with similar physical performance should also have similar embeddings. Specifically, for the leaf cell layout $i$ and $j$, the loss function is defined as follows:

$$\mathcal{L}_{ebd} = \sum_{(i,j) \in \mathcal{P}} \left( \text{sim}(e_i, e_j) - \text{sim}(\mathbf{m}_i, \mathbf{m}_j) \right)^2, \quad (4)$$

where $\mathbf{m}$ is a vector consisting of three physical indicators $[t, w, v]$, representing used track count, wirelength, and via

count, respectively. $\mathcal{P}$ is the universal set of leaf cell layout cases in the supervised training. $e$ is the embedding output by the proposed representation model. $\text{sim}(.)$ represents cosine similarity.

The leaf cell layout representation can be used to evaluate the similarity between layouts from the perspective of the topology of actual physical design. Therefore, we can use this representation model to measure the gap between the generated layout and the expert design results during the former dataset preparation and the advantage calculation in the GRPO stage.

### 3.2. PnR APIs Implementation

It is challenging for LLMs to generate PnR results directly because existing LLMs struggle to completely understand the physical relationships and to generate legal layouts. Therefore, in GenLeaf, we leverage the powerful code generation capabilities of LLM to perform leaf cell PnR using the generated scripts.

We implemented the APIs for leaf cell PnR. Since the number of cells in a leaf cell layout is relatively low, we also define the placement permutation and rotation of cells in the scripts, allowing the LLM to place or rotate them directly. As for routing, the channel routing algorithm (Kahng et al., 2011), a widely used greedy algorithm, has been implemented for LLM calls. The detailed implementation of the channel routing API is shown in Section A.2 of the appendix. With the PnR APIs, we establish a mapping between layout and scripts and allow LLMs to design leaf cell layouts by generating scripts.

### 3.3. BO-based Layout-Script Dataset Preparation

Since there have been many expert-designed leaf cell layouts but no corresponding scripts, they cannot be used directly to train the LLM. To solve this problem, we propose a Bayesian Optimization (BO) based dataset preparation method to align layouts and scripts, obtaining the scripts corresponding to the layouts, as shown in Figure 3. Specif-

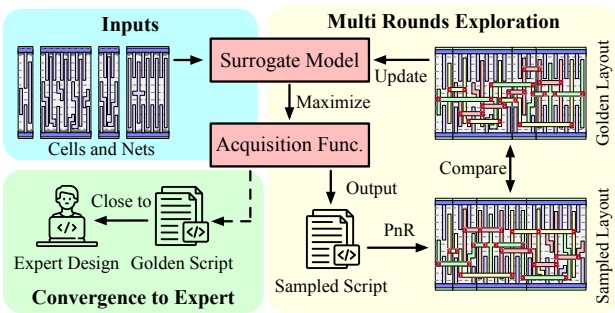

*Figure 3.* The BO-based layout-script dataset preparation. After multiple rounds of exploration, the surrogate model will eventually find the script close to the golden design.

ically, the objective of the BO process is to minimize the following function:

$$f(x) = (\alpha \cdot t + \beta \cdot w + \gamma \cdot v) - \text{sim}(e^b, e^g), \quad (5)$$

where $e^b$ represents the embeddings that correspond to the scripts that were searched by the BO process. And $e^g$ is the embedding of the golden layouts. $\alpha$, $\beta$, and $\gamma$ are the hyperparameters to balance the impact of the physical metrics and similarity. The first three terms are used to optimize the track count, wirelength, and via count. The last term is to allow BO to find the scripts that correspond as closely as possible to the original PnR results. $x$ ($x = [o, r]$) is the decision variable, consisting of the placement permutation $o$ and flip variable $r$.

First, we assume the objective function $f(x)$ follows a Gaussian Process (GP) prior (MacKay et al., 1998; Seeger, 2004), as shown in Equation (6). With the GP as the surrogate model, the objective function can be learned with a finite number of evaluations.

$$f(x) \sim \mathcal{GP}(m(x), k(x, x')), \quad (6)$$

where $m(x)$ is the mean function, that is, the prior expectation of the objective function $f(x)$ at the input point $x$. And $k(x, x')$ is the kernel function that defines the covariance between any two points $(x, f(x))$ and $(x', f(x'))$ as a prior assumption of the objective function. Given observed data $\mathcal{D}_n = \{(x_i, y_i)\}_{i=1}^n$, the posterior distribution at a new point $x$ is $f(x)|\mathcal{D}_n \sim \mathcal{N}(\mu_n(x), \sigma_n^2(x))$. Equation (7) and Equation (8) define $\mu$ and $\sigma$, respectively. $\mathbf{K}$ is the kernel matrix with $K_{ij} = k(x_i, x_j)$. $k(x)$ is $[k(x_1, x), \ldots, k(x_n, x)]^T$. $\sigma_\epsilon^2$ is the noise parameter to avoid singularities during matrix inversion.

$$\mu_n(x) = \mathbf{k}(x)^T (\mathbf{K} + \sigma_\epsilon^2 \mathbf{I})^{-1} \mathbf{y} \quad (7)$$

$$\sigma_n^2(x) = k(x, x) - \mathbf{k}(x)^T (\mathbf{K} + \sigma_\epsilon^2 \mathbf{I})^{-1} \mathbf{k}(x) \quad (8)$$

As for the acquisition function, we select the Expected Improvement (EI) (Jones et al., 1998; Zhan & Xing, 2020)

because it can balance exploration and exploitation. Specifically, the EI function is defined in Equation (9), where $\mathbb{E}$ represents the mathematical expectation. The reason why we need expectation $\mathbb{E}$ is that the prediction of $f(x)$ from GP is not a determined value, but a probability distribution.

$$\text{EI}(x) = \mathbb{E}[\max(f_{\text{best}} - f(x), 0)] \quad (9)$$

Notably, for the efficiency of BO, it is necessary to minimize illegal results. Therefore, we used the Lehmer code (Lehmer, 1960) on the permutation variable $o$ to ensure that each attempt results in a legal placement, as introduced in Section A.1 of the appendix. Meanwhile, $\mathcal{O}$ is the universal set of possible placement permutations after Lehmer encoding, as shown in Equation (10). The flip variable $\mathcal{R}$ determines whether the corresponding cells are flipped. R0 and MY separately represent: maintaining the default angle and flipping along the y-axis, as shown in Equation (10).

$$\mathcal{O} = \prod_{i=1}^n \{1, 2, \ldots, n - i\}, \quad \mathcal{R} = \{\text{R0}, \text{MY}\}^n \quad (10)$$

As defined in Equation (10), in a leaf cell placement case with $n$ cells, the solution space can be estimated as $n! \cdot 2^n$. Even for relatively small designs, the solution space remains huge. Therefore, we propose the two-stage specialization for GenLeaf to improve its performance in generating high-quality leaf cell layouts.

### 3.4. Specialized LLM-based Layout Generation

For stability and performance considerations, we propose a two-stage specialization method for LLM-based leaf cell layout generation, corresponding to the SFT and GRPO. Through this method, the performance of the generated layouts can be significantly enhanced.

#### 3.4.1. SFT STAGE

After preparing the layout-script dataset, we obtain all the scripts corresponding to the golden layouts. Then we perform Supervised Fine-tuning (SFT) for GenLeaf, as shown in Figure 4. The purpose of the SFT stage is to enable GenLeaf to gain a basic understanding of the calling order and methods of relevant PnR APIs, and to refer to the golden scripts generated from the golden layouts.

We created a generic template for importing relevant library functions and storing layout results, thereby improving the robustness of script generation. Therefore, the calculation of the loss function only needs to be limited to the design method selection and related parameters in the generated scripts, as follows:

$$\mathcal{L}_{SFT} = \frac{1}{N} \sum_{i=1}^N [-z_{i, y_i} + \log(\sum_{j=1}^V \exp(z_{i,j}))], \quad (11)$$

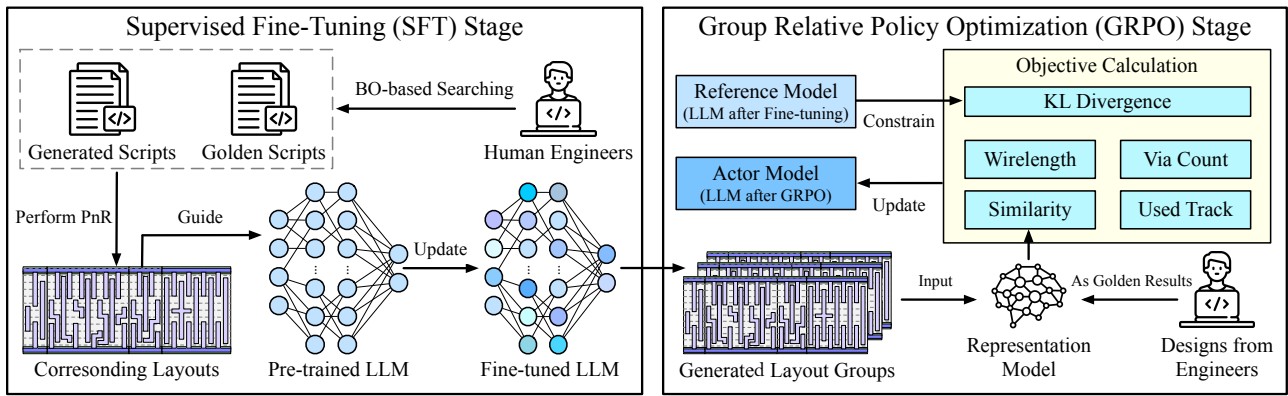

*Figure 4.* The two-stage specialization for GenLeaf, including SFT and GRPO stages.

where $z_{i,j}$ is the logit of the $i$-th token, and $z_{i,y_i}$ is the corresponding logit from the ground truth script. $V$ is the size of the vocabulary. In the loss function, the first term ensures that the correct tokens receive higher scores. The second term performs normalization on the logits, preventing the value of $z$ from increasing indefinitely, thereby ensuring the normal operation of SFT.

To improve efficiency, we used Low-Rank Adaptation (LoRA) (Hu et al., 2022), which significantly reduces the memory requirements by updating only a small number of parameters. LoRA also preserves the ability of the base model, avoiding catastrophic forgetting.

### 3.4.2. GRPO STAGE

The SFT stage helps GenLeaf establish the basic ability to build high-quality PnR scripts by calling the APIs, and the subsequent GRPO stage is based on the fine-tuned LLM and further improves the performance by aligning with expert preferences, as shown in Figure 4.

Specifically, the LLM agent generates multiple scripts under the same queries, and each of these scripts corresponds to a layout. These scripts are then organized into multiple groups. Since manually scoring the scripts is difficult to achieve and would introduce significant instability, GenLeaf applies the physical metrics and similarity to golden layouts to evaluate the relative advantage within groups. This method bridges the gap between automated algorithms and human engineers, thereby robustly improving layout quality.

GenLeaf first samples a group of queries and corresponding output scripts $\mathcal{S} = \{s_1, s_2, \ldots, s_n\}$ from the old model $\pi_{\theta_{old}}$, then optimizes the policy model by maximizing the following objective function:

$$\mathcal{J}(\theta) = \mathbb{E}[q \sim Q, \mathcal{S}_i \sim \pi_{\theta_{old}}(O|q)]$$
$$\left[\frac{1}{\mathcal{P} \cdot N} \sum_{i=1}^{\mathcal{P}} \sum_{j=1}^{N} \left[\min\left(I_j \hat{A}_j, \text{clip}(I_j, 1-\epsilon, 1+\epsilon)\hat{A}_j\right) - \kappa \cdot \mathbb{D}_{\text{KL}_j}\right]\right], \quad (12)$$

where $I_j$ denotes the probability ratio of the current model

$\pi_\theta$ to the old model $\pi_{\theta_{old}}$ in generating a specific script $s_j$:

$$I_j = \frac{\pi_\theta(s_j|q)}{\pi_{\theta_{old}}(s_j|q)}, \quad (13)$$

and clip(.) is a function to restrict to a specified range in a parameter update round. $\epsilon$ is the hyperparameter to set the min-max bound of the clip(.). $\mathbb{D}_{\text{KL}_j}$ is the Kullback-Leibler divergence, which integrates the unbiased estimator (Schulman, 2020) to guarantee its positive value, as shown in Equation (14). This term controls the constraint from the reference model $\pi_{\text{ref}}$ and prevents the updated model $\pi_\theta$ from deviating too much. In our implementation, the LLM after SFT serves as the reference model $\pi_{\text{ref}}$.

$$\mathbb{D}_{\text{KL}_j} = \frac{\pi_{\text{ref}}(s_j|q)}{\pi_\theta(s_j|q)} - \log \frac{\pi_{\text{ref}}(s_j|q)}{\pi_\theta(s_j|q)} - 1 \quad (14)$$

The relative advantage is calculated based on both the actual physical metrics and the similarity to the golden layouts. Specifically, the physical metrics include the used track count $t$, the wirelength $w$, and the via count $v$. As for the similarity to the golden layouts, we utilize the layout representation model. Different from the conventional settings in GRPO (Shao et al., 2024), in GenLeaf, the customized advantage $\hat{A}$ of script $s_j$ is that:

$$\hat{A}_j = (\alpha \cdot \frac{\mu_t - t_j}{\sigma_t} + \beta \cdot \frac{\mu_w - w_j}{\sigma_w} + \gamma \cdot \frac{\mu_v - v_j}{\sigma_v}) + \text{sim}(e_j, e_j^g) \quad (15)$$

In the first three terms, $\mu$ and $\sigma$ represent the average value and standard deviation of the three physical metrics, respectively. And the last term is the similarity between the embedding $e_j$ of the LLM-generated layout and the corresponding golden layout's embedding $e_j^g$. The embeddings are generated by the representation model introduced in Section 3.1.

Through the above-mentioned SFT and GRPO, GenLeaf can bridge the gap between LLM and expert designers, providing layouts that approximate the result of manual design.

# 4. Experiment

## 4.1. Experiment Settings

We implement GenLeaf in Python, PyTorch, and Hugging-Face Transformers for LLM inference and training. For the SFT and GRPO, we select Qwen3 (14B) (Yang et al., 2025) as the base model of GenLeaf. Cadence Virtuoso (Cadence, 2026) performs the design rule checks (DRC) and Layout Versus Schematic (LVS) verification. Our script-based generation ensures construction-correctness for most rules, and a lightweight post-processing step handles minor spacing violations to ensure legal final layouts. The unit of wire-length is $\mu m$. We run all the experiments on a Linux server equipped with 56 Intel Xeon CPU cores and 4 NVIDIA A100 GPUs, each with 80GB of memory.

In the SFT stage, the batch size is 4, and the learning rate is $2 \times 10^{-5}$. For the LoRA configuration, the rank is 16, the alpha is 32, the dropout value is 0.05, and the target modules include query, key, value, and output of attention mechanism components. The learning rate of GRPO is set to $1 \times 10^{-5}$, $\kappa$ is 0.1 and $\epsilon$ is 0.2. The temperature of GenLeaf is set to 0.03, and the maximum sequence length is 8192 to handle long-context script outputs. The hyperparameters $\alpha$, $\beta$, and $\gamma$ are analyzed in detail in Section C.3 of the appendix.

## 4.2. Dataset and Benchmarks

We collect datasets from actual industrial applications. 422 layout categories are used for training the representation model to ensure diversity. For the SFT and GRPO of Gen-Leaf, 272 and 156 expert-designed layout categories are used as training data, respectively. In our datasets, each category includes multiple layout variants. It is worth noting that despite the limited dataset size, the scripts reverse-generated by our high-precision BO framework provide a clean and dense "textbook" of design rules, enabling the LLM to master the syntax of layout generation efficiently without requiring massive-scale pre-training data, as shown in Section C.1 of the appendix. In the GRPO stage, it generates 20 scripts for each input, and each group consists of 5 scripts to calculate the in-group advantage. To verify the performance of GenLeaf in the real world, all layout benchmarks are from the industry frontier, as shown in Table 3.

## 4.3. Baselines

The experiment is mainly divided into two parts: layout representation and PnR design. The baselines in the experiment of representation are: 1) Circuit GNN (Yang et al., 2022): a customized GNN framework that fuses both topological and geometrical features for circuit layouts. 2) DeepLay-out (Zhao et al., 2025b): a general layout representation framework through mask strategies.

*Table 2.* Error comparison of prediction task on leaf cell layout between GenLeaf and other baseline representation methods.

| Model | MSE | MAE |
|---|---|---|
| GenLeaf (Ours) | 0.849 | 0.722 |
| Circuit GNN (Yang et al., 2022) | 1.287 | 0.815 |
| DeepLayout (Zhao et al., 2025b) | 1.324 | 0.969 |

We also select several LLMs to compare their performance with our GenLeaf, including Qwen2.5 (7B), Qwen3 (14B) (Qwen et al., 2025), Qwen3-Coder (30B) (Yang et al., 2025), and DeepSeek-R1 (14B) (DeepSeek-AI et al., 2025), which are locally deployed. Furthermore, we select two powerful commercial LLMs, GPT-4o (OpenAI et al., 2024) and Claude Sonnet 4 (Anthropic, 2024), as baseline models.

The baseline in the PnR design experiment is Golden Design. These layouts were designed by expert engineers, who manually completed the whole PnR process. All of them have been directly applied in actual VLSI designs and have shown satisfying performance.

## 4.4. Experiment Results

### 4.4.1. LAYOUT REPRESENTATION

Since the leaf cell layout representation is proposed to calculate the similarity between expert-designed and LLM-generated layouts, we first evaluate the performance of Gen-Leaf on layout representation. Specifically, we design a comprehensive prediction task as a downstream task on leaf cell layout. The used tracks count, via count, and wirelength form a vector of three elements to be predicted.

We compare the Mean Absolute Error (MAE) and Mean Squared Error (MSE) of GenLeaf with those of other baselines. As shown in Table 2, our GenLeaf outperforms the two state-of-the-art representation frameworks. Compared with Circuit GNN and DeepLayout, GenLeaf reduces the MSE by 34.0% and 35.9%, respectively. And the MAE of GenLeaf is also significantly lower than the baselines by 11.4% and 25.5%. The great performance on this downstream task demonstrates the effectiveness of the leaf cell layout representation of GenLeaf.

### 4.4.2. LEAF CELL LAYOUT DESIGN

**Overall Performance.** As shown in Table 3, we compare performance in generating leaf cell layouts between Gen-Leaf and human experts. Notably, in some complex cases with larger solution spaces, such as L4 and L10, GenLeaf can perform PnR with lower physical resources than the baseline. Specifically, compared with Golden Design, Gen-Leaf achieves a 4% reduction in wirelength with only a 1% increase in via count. As for the vital metrics, the track count, GenLeaf uses 48% fewer tracks, which means the

*Table 3.* Performance comparison of layout generation between GenLeaf and Golden Design. $w$, $v$, and $t$ represent wirelength, via count, and the used track, respectively.

| Case | Golden Design | | | GenLeaf (Ours) | | |
|------|--------|------|------|--------|------|------|
| | #Track | WL | #Via | #Track | WL | #Via |
| L1 | 7 | 29.91 | 18 | 3 | 30.66 | 18 |
| L2 | 7 | 42.01 | 21 | 3 | 43.09 | 21 |
| L3 | 8 | 54.03 | 24 | 3 | 53.72 | 24 |
| L4 | 7 | 66.29 | 27 | 3 | 64.83 | 27 |
| L5 | 6 | 29.80 | 25 | 5 | 32.22 | 22 |
| L6 | 2 | 32.19 | 11 | 2 | 33.11 | 11 |
| L7 | 13 | 46.22 | 30 | 7 | 43.71 | 32 |
| L8 | 4 | 9.07 | 4 | 3 | 10.84 | 9 |
| L9 | 3 | 49.10 | 8 | 2 | 36.85 | 8 |
| L10 | 5 | 66.51 | 27 | 3 | 61.12 | 27 |
| L11 | 9 | 15.41 | 16 | 5 | 20.89 | 15 |
| L12 | 12 | 26.93 | 26 | 5 | 23.80 | 28 |
| L13 | 13 | 62.99 | 31 | 8 | 62.97 | 23 |
| L14 | 8 | 36.82 | 19 | 3 | 28.90 | 27 |
| Avg. ratio | 1.00 | 1.00 | 1.00 | **0.52** | **0.96** | **1.01** |

*Table 4.* Performance comparison of layout generation between GenLeaf and other baseline LLMs.

| Model | Track | Wirelength | Via |
|-------|-------|------------|-----|
| **GenLeaf (Ours)** | **3.9** | **39.05** | **20.8** |
| Qwen2.5 (7B) | 6.3 | 63.71 | 36.1 |
| Qwen3 (14B) | 5.5 | 59.74 | 30.9 |
| DeepSeek-R1 (14B) | 5.6 | 60.83 | 30.5 |
| Qwen3-Coder (30B) | 5.1 | 52.70 | 26.9 |
| GPT-4o (API) | 4.8 | 47.50 | 24.3 |
| Claude Sonnet 4 (API) | 5.0 | 46.36 | 22.8 |

leaf cell layouts designed by GenLeaf can significantly save routing resources and meet specific design requirements. As for runtime, golden leaf cell layouts typically take several days to design by human engineers, whereas GenLeaf takes only 19.76s on average for inference.

The main reason why GenLeaf has such excellent performance is that it experienced SFT and GRPO. This specialization process not only helps GenLeaf reference the results and design preferences of human experts but also performs targeted optimizations based on physical metrics in leaf cell layout design. To further illustrate the superiority of GenLeaf, we also compare its performance with that of the traditional optimization method, **MILP+CRouter**, in Section C.2 of the appendix. The results demonstrate that GenLeaf achieves 6%, 12%, and 2% reductions in track usage, wirelength, and via count, respectively.

**LLM Implementations.** To further validate the training effectiveness, we compare GenLeaf with other baseline LLMs, including those deployed locally and those called by APIs. As shown in Table 4, our GenLeaf outperforms all baselines. including the powerful commercial models: GPT-4o and

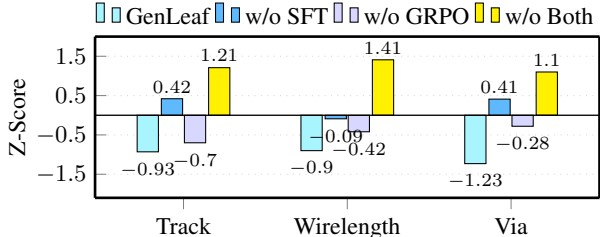

*Figure 5.* Ablation study of GenLeaf. The results are processed using z-score normalization, and higher scores indicate a greater importance of the removed part in improving the performance.

Claude Sonnet 4. Even compared to Claude Sonnet 4, GenLeaf still achieves reductions of 22% in used track count, 15.7% in wirelength, and 8.8% in via count. Compared to the base model, Qwen3 (14B), GenLeaf shows 29.1%, 34.6%, and 32.7% reductions in resource consumption for track, wirelength, and via, respectively.

**Ablation Study.** To further demonstrate the effectiveness of SFT and GRPO in GenLeaf, we conducted an ablation study for different configurations, as shown in Figure 5. Overall, without either of these two stages, the layout performance fails to be optimized, demonstrating their importance in improving it. It is notable that GenLeaf without SFT shows significantly worse across all three metrics. Specifically, without SFT, the used track count, wirelength, and via count of generated layouts increase by 30.9%, 21.7%, and 36.6%, respectively. This directly demonstrates the effectiveness and importance of SFT, helping GenLeaf understand the basic usage of PnR APIs and the direction for optimizing physical metrics. As for GenLeaf without GRPO, all three metrics also show significantly worse performance. The used track count, wirelength, and via count without GRPO show increases of 3.3%, 13.2%, and 20.0%, respectively. The results demonstrate the vital role of GRPO in optimizing the quality of layouts generated by GenLeaf.

## 5. Conclusion

This paper proposed GenLeaf, the first LLM-assisted framework for generating expert-level leaf cell layouts. For automated evaluation and dataset generation of leaf cell layouts, we first proposed a supervised, performance-aware embedding model to represent layouts and calculate their similarities. Because there are many expert-designed layouts but no corresponding scripts, we also proposed a BO-based dataset preparation method to align the layouts and scripts. After that, SFT is deployed to help GenLeaf learn the PnR APIs and perform initial optimization. Then, we utilize GRPO to further enhance the performance and meet the design preferences of human engineers. Experiment results show that GenLeaf can generate high-quality layouts that rival or exceed expert-designed golden designs.

## Impact Statement

This paper presents work whose goal is to advance the field of Electronic Design Automation and Artificial Intelligence. By automating the design of leaf cells, the fundamental building blocks of chips, GenLeaf has the potential to significantly accelerate the chip design cycle, reduce development costs, and improve the performance of integrated circuits. While this technology reduces the manual workload for layout engineers, it is designed to enhance, rather than replace, human expertise, allowing engineers to focus on higher-level architecture optimization. We do not foresee any immediate negative societal consequences or ethical concerns.

## Acknowledgement

This work is supported by the National Key Research and Development Program of China (No. 2023YFB4402900), the National Natural Science Foundation of China (No. 92573108, 62304197). This paper was conducted in the JC STEM Lab of Intelligent Design Automation funded by The Hong Kong Jockey Club Charities Trust.

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

# A. Implementation Details

## A.1. Lehmer Codec for BO-based Data Preparation

Section 3.3 briefly explains how we use the Lehmer code to ensure that the BO surrogate model can obtain legal placement results for each exploration of the placement location of cells. In this section, the Lehmer encoding and decoding are introduced in detail.

Algorithm 2 shows the Lehmer encoding process. The algorithm converts the Lehmer code to an integer as the regulatory factor for efficient BO exploration. The surrogate model only needs to update the regulatory factor to explore a unique placement setting. First, we initialize an empty sequence $\mathcal{O}$ to store the Lehmer code digits (line 2). Then, the algorithm iterates over each element $b_i$ in the placement permutation $\mathcal{B}$ where $i$ ranges from 1 to $n$ (line 3). For each $b_i$, the algorithm counts how many subsequent elements $b_j$ (where $j > i$) are smaller than $b_i$. This count represents the number of inversions to the right of $b_i$ (lines 5-9). After that, the count is stored as $o_i \in [0, n-i]$, the $i$-th digit of the Lehmer code (line 10). Then, the algorithm accumulates a weighted sum where each digit is multiplied by the factorial of its remaining positions (lines 15-17). Eventually, the final regulatory factor $k$ is returned, uniquely representing the permutation $\mathcal{B}$ with the intermediary Lehmer code sequence $\mathcal{O}$.

---

**Algorithm 2** Lehmer Encoding: Permutation to Regulatory Factor

---

**Require:** Permutation $\mathcal{B} = [b_1, b_2, \ldots, b_n]$ of elements $\{1, 2, \ldots, n\}$
**Ensure:** Integer representation $k \in [0, n! - 1]$

1: Initialize $\mathcal{O} \leftarrow \emptyset$                                                     // Initialize the sequence of Lehmer Code
2: **for** $i \leftarrow (1, n)$ **do**
3:     $\text{cnt} \leftarrow 0$
4:     **for** $j \leftarrow (i+1, n)$ **do**
5:         **if** $b_j < b_i$ **then**
6:             $\text{cnt} \leftarrow \text{cnt} + 1$                                   // Found a smaller element to the right
7:         **end if**
8:     **end for**
9:     $o_i \leftarrow \text{cnt}$                                         // $o_i \in [0, n-i]$ by construction
10: **end for**
11: $k \leftarrow 0$
12: **for** $i \leftarrow (1, n)$ **do**
13:     $k \leftarrow k + \mathcal{O}_i \times (n-i)!$                              // Accumulate weighted sum
14: **end for**
15: **return** $k$                                         // Unique integer in $[0, n! - 1]$ representing $\mathcal{O}$

---

Algorithm 3 decodes the regulatory factor $k$ into the corresponding permutation using the Lehmer code. First, the algorithm initializes an empty sequence $\mathcal{O}$ to store the Lehmer code digits (line 2). Then, for each $i$, it computes the $i$-th Lehmer digit by integer division and updates $k$ to the remainder $k \mod (n-i)!$ (lines 4-5). After that, the algorithm initializes a sorted list $S$ of available numbers and an empty sequence $\mathcal{B}$ for the permutation (lines 8-9). For each $i$, it selects the $(\mathcal{O}_i + 1)$-th smallest element from $S$ and appends it to $\mathcal{B}$ (line 12). After the traversal, the algorithm returns the permutation $\mathcal{B}$.

With the implementation of the proposed Lehmer codec, the surrogate model of BO can adjust the Lehmer code by simply changing the integer $k$, thereby altering the permutation $\mathcal{B}$ for each cell in the leaf cell layouts.

## A.2. Implementation of Channel Routing

The used channel routing method is shown in Algorithm 4. First, the algorithm builds a conflict graph where nets are connected by an edge if their horizontal spans overlap (lines 2-11). Specifically, for each pair of nets in the network, the algorithm checks if their spans overlap (line 5). If overlap exists, it adds bidirectional conflict edges between the nets (lines 6-7). As for multi-layer track assignment, the algorithm first sets an assignment flag (line 16). Then, it iterates through all layers and existing tracks to find a compatible solution (lines 17-26). Specifically, the algorithm first checks if a track is compatible, that is, there are no conflicts with existing nets in that track, and no direct overlap (line 19). If compatible, it assigns the net to that track and layer, updates the track and marks as assigned (lines 20-22). If no compatible track is found in any layer, the algorithm then creates a new track in layer $q$ by default.

---

**Algorithm 3** Lehmer Decoding: Regulatory Factor to Permutation

---

**Require:** Integer $k \in [0, n! - 1]$, permutation size $n$
**Ensure:** Permutation $\mathcal{B} = [b_1, b_2, \ldots, b_n]$

  1: Initialize $\mathcal{O} \leftarrow \emptyset$
  2: **for** $i \leftarrow (1, n)$ **do**
  3:    $\mathcal{O}_i \leftarrow \lfloor k/(n - i)! \rfloor$               // Extract the $i$-th Lehmer digit
  4:    $k \leftarrow k \mod (n - i)!$           // Remainder for next iteration
  5: **end for**
  6: Initialize available set $S \leftarrow \{1, 2, \ldots, n\}$         // Sorted list of candidates
  7: Initialize $\mathcal{B} \leftarrow \emptyset$
  8: **for** $i \leftarrow (1, n)$ **do**
  9:    $b_i \leftarrow S[\mathcal{O}_i + 1]$           // Select $(L_i + 1)$-th smallest from $S$
10:    Remove $b_i$ from $S$
11: **end for**
12: **return** $\mathcal{B}$                // Reconstructed permutation

---

**Algorithm 4** Channel Routing Algorithm

---

**Require:** Set of nets $\mathcal{N}$, each net $n$ has horizontal span $(x_{min}^n, x_{max}^n)$
**Ensure:** Track and layer assignment for each net

  1: **function** BuildConstraintGraph($\mathcal{N}$)
  2:    $\mathcal{C} \leftarrow \emptyset$          // Initialize constraint set (adjacency list)
  3:    **for** each pair $(n_i, n_j)$ in $\mathcal{N}$ **do**
  4:      **if** Overlap(span($n_i$), span($n_j$)) **then**
  5:        $\mathcal{C}[n_i] \leftarrow \mathcal{C}[n_i] \cup \{n_j\}$      // Add conflict edge
  6:        $\mathcal{C}[n_j] \leftarrow \mathcal{C}[n_j] \cup \{n_i\}$      // Symmetric relation
  7:      **end if**
  8:    **end for**
  9:    **return** $\mathcal{C}$         // Return the constructed constraint graph
10: **end function**
11:
12: **function** AssignTracks($\mathcal{N}, \mathcal{C}$)
13:    Sort $\mathcal{N}$ by $x_{min}$ in ascending order
14:    $\mathcal{L}[1, 2, ..., L_{max}] \leftarrow \emptyset$     // Initialize $L_{max}$ layers, each with empty track lists
15:    **for** each net $n$ in sorted $\mathcal{N}$ **do**
16:      $assigned \leftarrow$ false      // Flag for assignment success
17:      **for** $l \leftarrow 1$ to $L_{max}$ **do**
18:        **for** each track $T_k$ in $\mathcal{L}[l]$ **do**
19:          **if** $\forall s \in T_k : s.net \notin \mathcal{C}[n] \wedge \neg$ Overlap($s, n$) **then**
20:            $n.track \leftarrow k, n.layer \leftarrow l$     // Assign track and layer
21:            $T_k \leftarrow T_k \cup \{n\}$      // Add net to track
22:            $assigned \leftarrow$ true      // Mark as assigned
23:            **break**
24:          **end if**
25:        **end for**
26:      **end for**
27:      **if** not $assigned$ **then**
28:        Create new track $T_{new}$ in $\mathcal{L}[q]$     // Default to layer q for new track
29:        $n.track \leftarrow |\mathcal{L}[q]|, n.layer \leftarrow q$     // Assign to new track
30:      **end if**
31:    **end for**
32: **end function**

---

# B. Backgrounds

### B.1. Leaf Cell Physical Design

Physical design is a vital process in VLSI design. The quality of the leaf cell physical design directly affects the PPA (Power, Performance, and Area) performance of the final VLSI designs. This process can be divided into two parts: Placement and Routing. The objective of placement is to determine the locations and orientations of cells. All cells should be placed within the boundary of the layout, and no overlapping between cells is allowed. Routing is the process of determining the wiring required to connect all the nets in a given layout, so that the layout can run correctly. There may be multiple layers for routing, and the algorithm is allowed to use the via to route traces on another layer to avoid design violations. In leaf cell designs, all wire segments can only be on the track. Furthermore, track resources are extremely limited in the VLSI design. Therefore, in leaf cell design, the fewer tracks used, the more routing resources are available for subsequent design processes, and the better the overall performance.

### B.2. Circuit Layout Representation

Representation learning (Bengio et al., 2013) makes it easier to extract useful information and features of the data when building AI models. To better evaluate the design quality of leaf cell layouts, it is vital to represent them effectively. There are already some frameworks for circuit layout representation. CircuitGNN (Yang et al., 2022) integrates the topology and physical information into a graph to represent the circuit. DeepLayout (Zhao et al., 2025b) utilizes self-supervised training to perform circuit placement layout representation, so as to enhance the performance of multiple downstream tasks. However, because the scale of leaf cells is much smaller than that of VLSI, all the existing representation frameworks fail to provide proper embeddings for leaf cell layouts.

### B.3. Bayesian Optimization

Bayesian Optimization (BO) (Shahriari et al., 2016; Frazier, 2018) is a strategy for global optimization of black-box functions. In the context of EDA, we often deal with expensive-to-evaluate objectives (e.g., PnR simulations). BO constructs a probabilistic surrogate model, typically a Gaussian Process (GP), to approximate the objective function $f(x)$. It uses an acquisition function to decide where to sample next, balancing exploration (sampling high-uncertainty areas) and exploitation (sampling low-objective-value areas). In GenLeaf, BO is utilized to search for the optimal PnR scripts that result in layouts matching the target embeddings, enabling specialization of GenLeaf.

### B.4. Group Relative Policy Optimization

There are several methods for training the LLM agent to align with human preferences, namely Reinforcement Learning from Human Feedback (RLHF) (Christiano et al., 2017; Ziegler et al., 2019). Among them, the most influential method is Proximal Policy Optimization (PPO) (Schulman et al., 2017). However, PPO requires training a value network as a critic to precisely estimate the reward function. In some complicated tasks, such as circuit physical design, achieving this is exceptionally challenging.

Group Relative Policy Optimization (GRPO) (Shao et al., 2024) is an effective and straightforward method to perform RLHF (Christiano et al., 2017; Ziegler et al., 2019). It is a policy gradient method that extends PPO by incorporating group-wise advantage normalization techniques to improve the parameter update process. Regarding the leaf cell layout design task, assigning scores to the results is unstable and difficult. But determining their relative ranks is much easier. Therefore, implementing GRPO in leaf cell layout design is a viable approach that could enable agents to learn from human expertise and generate expert-level layouts.

# C. Supplementary Experiment

### C.1. Result of BO-based Data Preparation

As introduced in Section 3.3, there are only golden layouts, not their corresponding scripts, that can be collected. Therefore, GenLeaf utilizes BO to search for the scripts whose corresponding layouts are similar to the golden designs. In our experiment, each script is searched for 400 iterations to be as close as possible to the golden result. Specifically, under the three metrics: the used track count, wirelength, and via count, the scripts searched by BO have an average difference of only 3.63% from the golden results. The result indicates that they can be used as golden scripts to train GenLeaf and the

*Table 5.* Performance comparison of layout generation between GenLeaf and MILP + CRouter.

| Method | Metric | L1 | L2 | L3 | L4 | L5 | L6 | L7 | L8 | L9 | L10 | L11 | L12 | L13 | L14 | Ratio |
|---|---|---|---|---|---|---|---|---|---|---|---|---|---|---|---|---|
| MILP + CRouter | #Track | 3 | 3 | 3 | 3 | 4 | 3 | 6 | 3 | 2 | 4 | 8 | 5 | 8 | 3 | 1.00 |
| | WL | 30.83 | 43.75 | 56.67 | 69.60 | 27.48 | 31.72 | 57.09 | 10.18 | 57.16 | 79.12 | 15.85 | 32.20 | 70.87 | 35.34 | 1.00 |
| | #Via | 18 | 21 | 24 | 27 | 22 | 11 | 32 | 9 | 8 | 27 | 16 | 28 | 23 | 31 | 1.00 |
| **GenLeaf (Ours)** | #Track | 3 | 3 | 3 | 3 | 5 | 2 | 7 | 3 | 2 | 3 | 5 | 5 | 8 | 3 | **0.94** |
| | WL | 30.66 | 43.09 | 53.72 | 64.83 | 32.22 | 33.11 | 43.71 | 10.84 | 36.85 | 61.12 | 20.89 | 23.80 | 62.97 | 28.90 | **0.88** |
| | #Via | 18 | 21 | 24 | 27 | 22 | 11 | 32 | 9 | 8 | 27 | 15 | 28 | 23 | 27 | **0.98** |

effectiveness of our proposed BO-based data preparation pipeline.

## C.2. Comparison with Optimization Method

To demonstrate the effectiveness of GenLeaf, we also design a traditional optimization-based method as a baseline: **MILP+CRouter**: We implemented the MILP-based placement. And the objective is shown in Equation (16):

$$\min \left( \sum_{k=1}^{|E|} \text{HPWL}_k(\mathcal{O}, \mathcal{R}) + \max_{p \in \text{Pin}} (\sum_{k=1}^{|E|} C_{p,k}(\mathcal{O}, \mathcal{R})) \right), \tag{16}$$

where $C_{p,k}$ represents the number of nets covering pin $p$. The constraints include no overlap between cells, and all cells must be placed within the boundary. The formulation aims to minimize the Half Perimeter Wirelength (HPWL) and the number of used tracks. We use Gurobi (Gurobi, 2026) to implement the optimization for the placement process. And CRouter is implemented using a greedy channel routing algorithm (Kahng et al., 2011), as shown in Algorithm 4.

The comparison result between GenLeaf and MILP + CRouter is shown in Table 5. Notably, GenLeaf demonstrates significantly better performance across all three metrics. Specifically, compared with MILP + CRouter, our GenLeaf saves 6% track, 2% via, and 12% wirelength, which can significantly improve the overall performance of VLSI. The reason why GenLeaf is overall superior to the traditional method is that it can learn from the design experience of experts, and explore possible better solutions through the SFT and GRPO stages.

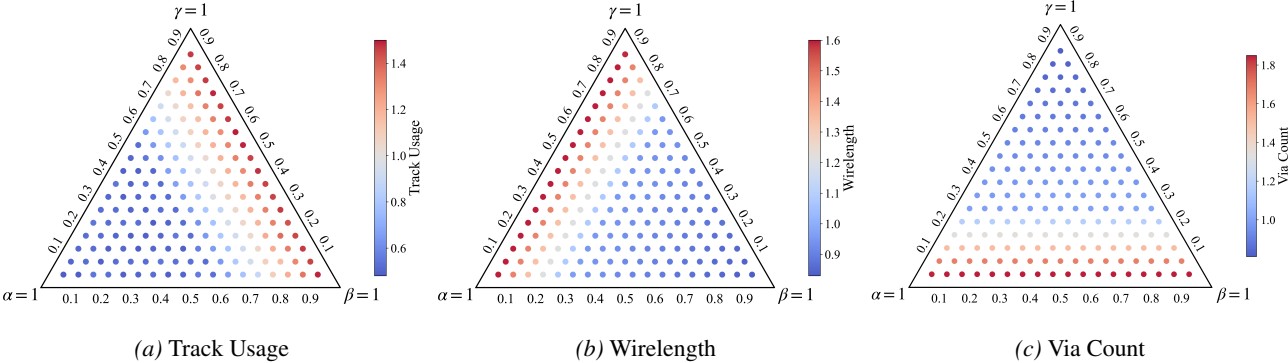

*(a)* Track Usage      *(b)* Wirelength      *(c)* Via Count

*Figure 6.* Hyperparameter analysis. All the values are the ratios of GenLeaf to Golden Design. The three subfigures show the relationships between $\alpha$, $\beta$, and $\gamma$ and the corresponding physical metrics.

## C.3. Hyperparameter Analysis

As introduced in Section 3, $\alpha$, $\beta$, and $\gamma$ are used to balance the three physical metrics: the used track count, wirelength, and via count while calculating the objective function in BO (Equation (5)) and the advantage function in GRPO (Equation (15)). In all the experiments in Section 4, $\alpha$, $\beta$, and $\gamma$ are set to 0.4, 0.3, and 0.3, respectively, since the used track count is a relatively more important metric in the leaf cell design process. To further analyze these hyperparameters and understand their respective roles, we vary their values within specified ranges in the experiment.

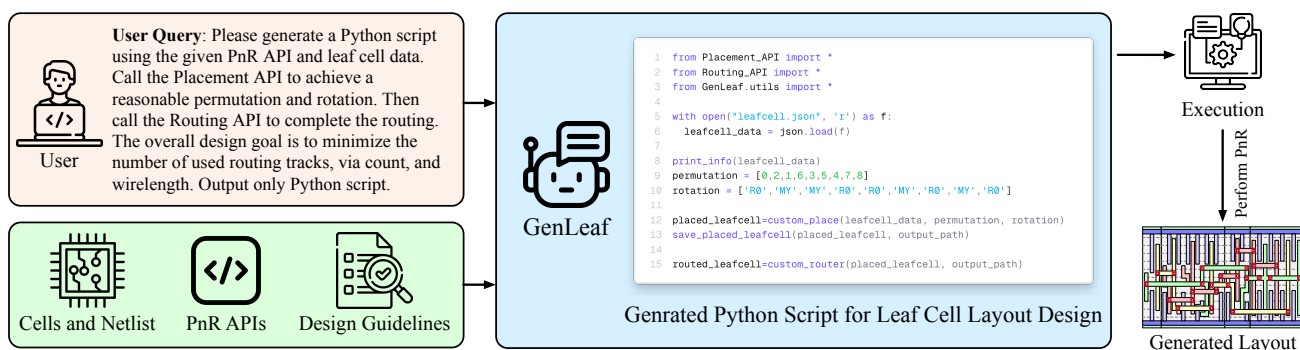

*Figure 7.* Case study of GenLeaf, which generates scripts to design leaf cell layouts.

As shown in Figure 6, it is clear that the advantage becomes more pronounced as the parameters corresponding to a metric increase. However, because $\alpha$, $\beta$, and $\gamma$ are inversely related, the performance of the other two metrics typically decreases to varying degrees. Since $\alpha$, $\beta$, and $\gamma$ represent the importance of three metrics, and these metrics may have conflicting relationships, adjusting their values can change the direction of optimization.

## D. Case Study

Figure 7 shows a case study of GenLeaf. The input is the user query, PnR APIs, design guidelines, and the information about the cells and netlist of the leaf cell to be designed. Then, GenLeaf inferences and generates Python scripts for leaf cell PnR design. With the help of the PnR engine, this code can be executed and output the whole leaf cell layout.

Compared to letting LLM generate some formats representing placement or routing results, code generation is a more efficient and robust method for PnR design. On the one hand, pre-trained LLMs already possess powerful capabilities for script generation and comprehension. On the other hand, as a medium directly corresponding to the layout, converting PnR to a script generation task can improve the efficiency and stability of LLM specification.

