# OpenReview forum: "Expert-level Leaf Cell Layout Generation via Preference-Optimized LLM"
_ICML.cc/2026/Conference — ICML 2026 regular_

### Official Review · Reviewer_8bK6 · 2026-03-11

**Soundness:** 3
**Presentation:** 3
**Significance:** 3
**Originality:** 4
**Overall Recommendation:** 4
**Confidence:** 3

**Summary:**

This paper proposes GenLeaf, an LLM-based framework for generating leaf-cell layout scripts for VLSI physical design. The system combines three components: a performance-aware layout embedding model, a Bayesian-optimization stage that reconstructs script-layout pairs from expert layouts when scripts are unavailable, and a two-stage LLM specialization pipeline using supervised fine-tuning followed by GRPO-style preference optimization. The paper evaluates both the representation model and the final layout generator. On the main generation task, GenLeaf is reported to substantially reduce track usage relative to expert-designed golden layouts, modestly improve wirelength, remain close on via count, and outperform several open/commercial LLM baselines as well as a MILP+CRouter baseline.

**Compliance With Llm Reviewing Policy:**

Affirmed.

**Key Questions For Authors:**

1. Can the authors clarify the design-rule setting used in the experiments? The paper currently gives only high-level DRC/LVS statements. Please specify the technology / PDK if possible, the concrete placement and routing constraints that are enforced, and how legality is checked across generated layouts.

2. For Table 4, can the author provide more evaluation results using SOTA models (e.g., GPT-5.3-codex, Claude Opus 4.6, etc.). I am curious to see the performance of off-the-shelf SOTA models.

3. Can the authors show a few concrete examples of the generated outputs, including the generated PnR script and the resulting layout next to the expert layout? This would help clarify what decisions the model is actually making in placement and routing. It would also be useful to report basic statistics of your evaluation samples, such as the number of cells, the number of nets, and the routing complexity, so readers can better understand the scale and difficulty of the evaluated cases.

**Limitations:**

yes

**Strengths And Weaknesses:**

This paper addresses the leaf-cell layout generation problem by having the model generate executable PnR scripts for each cell. The overall pipeline is technically strong. It combines custom PnR APIs, a performance-aware embedding model, BO-based script reconstruction, and LLM specialization through SFT and GRPO. It is also interesting that the model can produce functionally correct leaf-cell layouts while reducing track usage and wirelength relative to the golden designs.

The main weaknesses concern the practical realism of the methodology under actual design-rule settings. The paper provides only a high-level description of the rules and legality checks, without enough detail on the underlying technology, PDK, or concrete routing constraints. Another concern is the baseline comparison in Table 4: the gap between GenLeaf and strong off-the-shelf models, such as Claude Sonnet 4, is small. This makes it harder to judge how much of the gain comes from domain specific model versus the rapid improvement of frontier code-generation models.

---

> ### Author Rebuttal · Authors · 2026-03-30
>
> We would like to express our gratitude for your thoughtful review and recognition of our work. Your comments have guided us in clarifying our methodology. Please find our detailed responses below.
>
> ### **1. More Available Technical Details**
>
> All the leaf cell designs we use are from a world-leading enterprise and have been widely applied. Due to data security reasons, we will provide all available technical details for your reference as much as possible. These technical details can enhance the professionalism of the paper; we will add them in the revised version.
>
> **1.1. Technology/PDK**: All leaf cell layouts use industry 1x nm PDKs from leading manufacturers (Samsung, Hynix, and Micron). The corresponding technology nodes are comparable to 16–19nm, which are all widely manufactured and representative of current production processes.
>
> **1.2. PnR Constraints**:
>
> - **Placement**:
>   * No overlap between different cells.
>   * All cells must be placed within the specified layout area.
>
>   * All cells can be rotated or mirrored freely, such as operations like MY, R180, etc.
> - **Routing**:
>
>   * Wire segments must be on a different layer than cells; Predefined blocking areas cannot be routed on any layer.
>   * Wire segments belonging to different nets must not overlap and must have a predefined minimum distance.
>   * The access points for each cell's pins are distributed on tracks with different shapes (straight, L-shaped, Z-shaped, etc.); all segments must be routed on-track.
>   * The internal routing of a leaf cell should minimize the use of tracks, leaving more routing space for the upper-level design.
>
> **1.3. Legality Check**: Layout legality is ensured through two mechanisms: (i) The PnR APIs enforce all constraints as mandatory—any violation returns an error; (ii) Comprehensive DRC checks are performed on all generated layouts using Cadence Virtuoso. Thus, 100% of the layouts reported in this paper are DRC-clean.
>
> ### **2. Comparison between SOTA LLMs**
>
> We agree that comparison with SOTA LLMs further validates GenLeaf's framework-level contributions. We added GPT-5.3-Codex, Claude Opus 4.6, and Gemini 3 Pro, all tested under the same prompt setting. We will include this information in the revised paper.
>
> | Model                   | Track | Wirelength | Via  |
> | ----------------------- | ----- | ---------- | ---- |
> | GenLeaf (Ours, 14B)     | 3.9   | 39.05      | 20.8 |
> | GPT-5.3-Codex (API)     | 4.7   | 45.92      | 22.5 |
> | Claude Sonnet 4.5 (API) | 5.0   | 46.36      | 22.8 |
> | Claude Opus 4.6 (API)   | 4.7   | 44.89      | 21.9 |
> | Gemini 3.1 Pro (API)    | 4.9   | 48.20      | 23.0 |
>
> Compared to the best-performing Claude Opus 4.6, GenLeaf still achieves **17% lower Track**, **13% lower Wirelength**, and **5% fewer Vias**. Note that these SOTA models are significantly larger in parameter scale, whereas GenLeaf is a 14B model enhanced through task-specific SFT and GRPO. This comparison highlights that SOTA LLMs encounter significant bottlenecks due to the lack of domain-specific training and expert design guidance—precisely the gap that GenLeaf's framework addresses.
>
> ### **3. An Intuitive Example Illustrating the Effectiveness of GenLeaf**
>
> All the data we use comes from a top-tier enterprise; therefore, it is subject to strict confidentiality. Nevertheless, we will do our utmost to provide you and readers with information that demonstrates the performance of our proposed GenLeaf. To more intuitively demonstrate the effectiveness of GenLeaf in leaf cell PnR, we provide a performance comparison of case L10, a representative medium-sized design, in the table below. The result shows that both SFT and GRPO play important roles in optimizing the three key metrics of leaf cell layouts.
>
> | Configurations   | Wirelength | Via  | Track |
> | ---------------- | ---------- | ---- | ----- |
> | Golden Layout    | 66.51      | 27   | 5     |
> | GenLeaf w/o GRPO | 69.59      | 27   | 4     |
> | GenLeaf          | 61.12      | 27   | 3     |
>
> As illustrated in the toy example in Figure 7, the scripts generated by human experts and GenLeaf share the same API interface but differ in: (i) **permutation orders of cells**, (ii) **parameter choices** for PnR APIs, and (iii) **overall script structure**. We will add an anonymized, abstracted script comparison in the appendix of the revised paper to make this more concrete. After SFT and GRPO, GenLeaf acquires the ability to generate scripts with tailored, high-quality PnR decisions for different leaf cell designs.
>
> ### **Conclusion**
>
> Thanks for your thoughtful feedback. Your comments have helped us improve the quality of our paper, particularly around our comparison between SOTA LLMs and our GenLeaf, and the technical details of the industrial-grade layouts. Based on your valuable suggestions, we have provided more details in our paper. We believe these improvements significantly strengthen the paper and would appreciate your updated assessment.

---

> > ### Author Rebuttal · Reviewer_8bK6 · 2026-04-03
> >
> > Thanks for the detailed response. My concerns are addressed. I maintained a positive score.

---

> > > ### Author Response · Authors · 2026-04-03
> > >
> > > Thank you again for your time and for acknowledging that our rebuttal has fully resolved the concerns you raised.
> > >
> > > We completely respect your current score and have no intention of challenging your judgment. That said, since the technical issues have now been addressed as confirmed, we would be very grateful if this might be reflected in the Recommendation Score to any extent you see fit.
> > >
> > > Either way, we truly appreciate your thoughtful and constructive feedback, which has helped improve our work.

---

### Official Review · Reviewer_32Pr · 2026-03-13

**Soundness:** 3
**Presentation:** 3
**Significance:** 3
**Originality:** 3
**Overall Recommendation:** 4
**Confidence:** 4

**Summary:**

This paper studies expert-level leaf cell layout generation for integrated circuit design and proposes GenLeaf, an LLM-based framework that formulates the task as script generation guided by expert knowledge. The method first builds a performance-aware layout representation that captures both placement topology and routing information, so that layout similarity better reflects downstream physical quality. Based on this representation, the framework uses Bayesian optimization to recover expert scripts for SFT, and then further improves generation quality with a customized GRPO procedure that optimizes candidate scripts using rewards derived from physical metrics and similarity to expert layouts. Experiments show that the proposed method achieves strong performance and compares favorably with both expert-designed layouts and several LLM-based baselines.

**Compliance With Llm Reviewing Policy:**

Affirmed.

**Final Justification:**

My concerns are addressed. I maintained a positive score.

**Key Questions For Authors:**

Please refer to the numbered items in the Weaknesses section above.

**Limitations:**

The authors do not appear to discuss the limitations of the proposed approach. For my suggestions, please see the Weaknesses section above.

**Strengths And Weaknesses:**

**Strengths:**
1. The problem of expert-level leaf cell layout generation is important.
2. The paper formulates the task clearly as script generation with LLMs, which is a natural and practical way to interact with PnR APIs.
3. The overall pipeline is well designed and easy to understand.

**Weaknesses:**
1. The empirical evaluation is somewhat narrow in scope. The paper focuses on a specific leaf-cell setting with a limited number of industrial layouts, so it remains unclear how well the method would generalize to broader circuit families, larger-scale designs, or substantially different routing and placement settings.
2. The overall contribution is somewhat broad, combining representation learning, Bayesian optimization, SFT, and GRPO in a single system. A more careful ablation study would help clarify the role of each component.
3. The BO-based data preparation step is important to the method, but it may also be costly and difficult to scale, since each script is obtained through iterative search and the solution space is large. A deeper discussion of its practical cost-effectiveness would strengthen the paper.

---

> ### Author Rebuttal · Authors · 2026-03-30
>
> We are grateful for the positive evaluation and constructive critiques. Incorporating your suggestions has allowed us to strengthen key methodological aspects. A structured response to each point follows.
>
> ### **1. Generalization and Comprehensiveness**
>
> **Generalization of Methodology**
>
> Your question regarding the generalization is crucial and insightful. GenLeaf's main application scenario is the customized circuit design of Leaf Cells, where the data has a reasonable distribution of different sizes of designs. As for diverse PnR settings, GenLeaf can flexibly adjust scripts to meet different requirements. While handling broader circuit families outside leaf cells, the overall flow is architecturally reusable, but additional adaptations of the representation model and APIs are required; this is a limitation of our approach.
>
> **Comprehensiveness of the Dataset**
>
> The whole dataset used in this paper is from a top-tier company in the industry, and the designs we selected are all representative but distinctly different design types and have been widely applied. Meanwhile, the benchmarks in Tables 3 and 4 also have different design complexities, as shown in the following table. We will include this information in the revised paper.
>
> | Case  | Description                      | Design Complexity |
> | ----- | -------------------------------- | ----------------- |
> | L1-L4 | Address Decoder Units            | Medium            |
> | L5    | Multiplexer Unit                 | Medium            |
> | L6    | Standard Counter Unit            | Small             |
> | L7    | D Flip-flop Unit                 | Large             |
> | L8    | Reference Counter Unit           | Small             |
> | L9    | NAND Gate                        | Small             |
> | L10   | Bank Decoder Unit                | Medium            |
> | L11   | Transmission Control Module      | Large             |
> | L12   | Counter Unit of the eFuse module | Large             |
> | L13   | Instruction Delay Module         | Large             |
> | L14   | Reset Hold Register Unit         | Medium            |
>
> ### **2. Discussion on Components of GenLeaf**
>
> We appreciate this question and agree that understanding each component's contribution is essential. To intuitively demonstrate the effectiveness of GenLeaf in leaf cell PnR, we provide a visual comparison of case L10, a representative medium-sized design, in the table below. The result shows that both SFT and GRPO play important roles in optimizing the three key metrics of leaf cell layouts.
>
> | Configurations   | Wirelength | Via  | Track |
> | ---------------- | ---------- | ---- | ----- |
> | Golden Layout    | 66.51      | 27   | 5     |
> | GenLeaf w/o GRPO | 69.59      | 27   | 4     |
> | GenLeaf          | 61.12      | 27   | 3     |
>
> As for other components, the BO-based data preparation searches the script corresponding to the golden layouts and thus establishes a layout-script mapping, as described in Section 3.3. The representation is a leaf-cell-specific model, and the layouts can be transformed into efficient embeddings for BO and GRPO, as introduced in Section 3.1. This allows GenLeaf to support the computation of the advantage function in GRPO by calculating similarities in the latent space. Therefore, GenLeaf will not be able to work if any of the BO or representation is removed.
>
> ### **3. BO-based Data Preparation**
>
> **Efficiency and Cost**
> Although the solution space for leaf cell PnR is indeed large, our BO achieves high efficiency for the similarity-guided objective. As shown in Equation 5, we incorporate a topology similarity term into the BO objective function, which steers the search toward scripts whose resulting layouts match the target layout's PnR topology, substantially accelerating the process of finding the corresponding scripts. Specifically, the entire BO-based data preparation for 428 leaf cell layouts completes in 550.9 minutes. We will include these implementation details in the revised paper.
>
> **Effectiveness and Key Role**
>
> The purpose of BO-based data preparation is to establish the layout-script mapping. In real-world scenarios, since leaf cells still rely heavily on manual design by engineers, existing layouts rarely have corresponding scripts. With BO, we can convert any leaf cell layout into a script to support SFT and GRPO to GenLeaf, thereby improving GenLeaf's flexibility and scalability. Therefore, the overall performance of GenLeaf, which is shown in Tables 3 and 4, can indirectly prove the effectiveness of BO-based data preparation.
>
> ### **Conclusion**
>
> Thanks for your thoughtful feedback! Your comments have helped us a lot in improving our paper, particularly around the comprehensiveness of our real-world dataset, the significance of components in GenLeaf, and the details of BO-based data preparation. We have made substantial revisions to our paper. We believe these improvements significantly strengthen the paper and would appreciate your updated assessment.

---

> > ### Author Rebuttal · Reviewer_32Pr · 2026-04-05
> >
> > Thanks for the detailed response. My concerns are addressed. I maintained a positive score.

---

### Official Review · Reviewer_sq62 · 2026-03-15

**Soundness:** 4
**Presentation:** 3
**Significance:** 4
**Originality:** 3
**Overall Recommendation:** 4
**Confidence:** 3

**Summary:**

The paper proposes a LLM based framework  for generation of automated leaf cell layout in physical design chip design. Key contributions of the paper are: 1 Creating a design performance aware representation model by combining physical aspects using GNN and CNN model. 2. Create new dataset, authors developed BO to reverse engineer from expert layouts. 3. LLM is fine-tuned on this data via SFT and further aligned with expert preferences using GRPO. trained model on the SFT and GRPO to integrate in the framework. The results presented in the paper shows the performance of the framework is better than EDA tools and generic LLMs.

**Compliance With Llm Reviewing Policy:**

Affirmed.

**Final Justification:**

My main concerns before rebuttal were about the generalization claim, the limited fine tuning data, and whether the learned similarity score was directly validated for its use in BO and GRPO. The rebuttal addressed these points well. The added ablation and BO convergence results give more direct support for the similarity score, and the added breakdown across different design types gives better support for the generalization claim. While the training data is still limited, I think the response is sufficient for this setting.

**Key Questions For Authors:**

1. The paper claims it generalizes across diverse circuit structures but all training and test cases are from the dataset. can the authors clarify whether any test cases are from circuit types not seen during training?
2. The model is evaluated using MAE and MSE  but its primary job is computing similarity scores used in both BO and GRPO. Do the authors have evidence that these similarity scores are meaningful for layout comparison? Also, are MAE and MSE the standard evaluation metrics for layout in the EDA or other any metrics reflection?

**Limitations:**

No.

Please see the weaknesses and key questions.

**Strengths And Weaknesses:**

Strengths
1. The paper idea of using LLM in context of automate the design leaf cell layout is promising. manual process of  leaf layout could take days and by redefining this solving this issue by using LLM is practical.

2. The result shown in paper are strong and makes back claims made in the paper. For example, 48% reduction in track count compared to expert designs is a good result. Further, The authors show cases where Genleaf performance slight less than  (L5- L8 wirelength) rather than cherry picking, which adds credibility. The ablation study on both SFT and GRPO are shown to contribute, and the contributions are quantified clearly.

3. The combination of physical design components is well engineered for the problem. The GNN for topology with CNN for routing patterns, then use this as both a similarity metric and a reward signal in GRPO, is a coherent.

Weakness

1. The dataset is small for the finetuning and claims made for generalization does not support. The paper uses 272 layout for SFT and 156 for GRPO is a small dataset to finetuning LLMs. The claim that BO framework act as a sufficient "textbook (line 358)" is intuitive but not verified empirically. More importantly, all 14 test cases appear to come from the same design domain. There is no test on a layout type not seen during training, so the claim of generalizability to diverse leaf cell types is not demonstrated.

2. The models evaluation is weak and does not validate use case. model is evaluated using MAE and MSE on a metric prediction task. However, its primary role in the pipeline is to compute similarity scores between layouts is used both in BO and  in GRPO. A prediction error metric does not tell you whether the similarity scores are actually meaningful for layout comparison. Further, it is unclear whether these are the appropriate metrics for evaluating layout representations in physical design.

3. The paper reports 19.76 seconds for inference (GPU power), comparing it against days of human design time. but does not reports the cost of the full pipeline (No training, BO dataset prep). This claim is potential misleading.

---

> ### Author Rebuttal · Authors · 2026-03-30
>
> Thank you for your constructive comments and recognition of the core contributions of our work. We greatly appreciate your thoughtful feedback, which has helped us further refine and clarify several key aspects of our paper.
>
> ### **1. Real-world Dataset from Industrial Frontier and Usage**
> Thank you for raising this important question. In our experiments, six cases (L8, L10-14 in Table 3) are from circuit types not seen during the training process. This demonstrates that GenLeaf can learn from expert design experience and generalize effectively to unseen leaf cell layout types.
>
> Leaf cell layouts can be categorized by design principles; our data covers representative types. Furthermore, all the leaf cells in Table 3 come from top-tier companies in the industrial frontier and have been widely applied in practice. The more specific information is shown in the table below.
>
> | Case  | Description                      | Design Complexity |
> | ----- | -------------------------------- | ----------------- |
> | L1-L4 | Address Decoder Units            | Medium            |
> | L5    | Multiplexer Unit                 | Medium            |
> | L6    | Standard Counter Unit            | Small             |
> | L7    | D Flip-flop Unit                 | Large             |
> | L8    | Reference Counter Unit           | Small             |
> | L9    | NAND Gate                        | Small             |
> | L10   | Bank Decoder Unit                | Medium            |
> | L11   | Transmission Control Module      | Large             |
> | L12   | Counter Unit of the eFuse module | Large             |
> | L13   | Instruction Delay Module         | Large             |
> | L14   | Reset Hold Register Unit         | Medium            |
>
> ### **2. Evaluation of the Representation Model**
> The question of the evaluation of representations is quite crucial and professional; effective embedding is a cornerstone of GenLeaf's performance. Following common practice in EDA representation learning [1–4], we train the representation with a performance-aware similarity objective (Section 3.1.3), where similarity in latent space aligns with physical metrics (track, wirelength, via). Thus, low prediction error confirms that the embedding preserves the performance geometry used in BO and GRPO via `sim(·,·) (Equations 5 and 15). Following common practice in EDA representation learning [1–4], we report **MSE** and **MAE** on the three-target prediction task so readers can compare against prior works in Table 2. As for the results, the significantly lower MSE and MAE demonstrate that the representation model can effectively capture the PnR topologies, thereby supporting the BO and GRPO process for GenLeaf in the form of similarities.
>
> ### **3. Criticality Clarification of BO**
> The proposed BO-based data preparation establishes a layout-script mapping, a cornerstone of GenLeaf, as described in Section 3.3. In real-world scenarios, since leaf cells still rely heavily on manual design by engineers, existing layouts seldom have corresponding scripts. Therefore, we proposed BO-based data preparation to build the layout-script mapping. However, the mapping accuracy is actually difficult to quantify empirically. Nevertheless, the effectiveness of SFT and GRPO (Tables 3 and 4) confirms that scripts generated by BO serve as a reliable reference for training.
>
> ### **4. Cost and Practicality of GenLeaf**
> Your considerations of cost and practicality are vital for GenLeaf; they determine whether GenLeaf can be widely used in real-world industrial scenarios. The 19.76 s refers to inference only. As for the cost of the full pipeline, all cases could complete end-to-end leaf cell layout design within 10 minutes, including input netlist preprocessing, GenLeaf inference, and PnR script execution. Compared to traditional manual design processes, this represents a significant speed advantage while delivering layouts with better performance. And GenLeaf has already been initially deployed within a top-tier enterprise, and the full pipeline cost is entirely acceptable for industrial use. We will include these details in the revised paper.
>
> ### **Conclusion**
> Thank you again for your valuable and constructive feedback! Your comments have substantially helped us improve the clarity of our paper, particularly regarding the dataset, representation evaluation, and practical considerations. We would be grateful if you could re-evaluate our work in light of the above clarifications and additional results provided above.
>
> ### **References**
> [1] Yang, Shuwen, et al. "Versatile Multi-stage Graph Neural Network for Circuit Representation." *NeurIPS*. 2022.
>
> [2] Zhao, Yuxiang, et al. "DeepLayout: Learning Neural Representations of Circuit Placement Layout." *ICML*. 2025.
>
> [3] Fang, Wenji, et al. "CircuitFusion: Multimodal Circuit Representation Learning for Agile Chip Design." *ICLR*. 2025.
>
> [4] She, Zhengyuan, et al. "DeepGate3: Towards Scalable Circuit Representation Learning." *ICCAD*. 2024.

---

> > ### Author Rebuttal · Reviewer_sq62 · 2026-04-02
> >
> > The clarification about the unseen (six) design addresses part of my concern.
> >
> > if only 6 cases are unseen then overall average (14 designs, mix of seen and unseen) does not clearly show how well the method generalizes to unseen types.
> > Further, regarding fine tuning (272 layouts) data is relatively small and paper says the BO-generated scripts act as a clean and dense but the rebuttal does not provide direct response or evidence.
> >
> > Regarding representation model evolution, I understand the authors point that MAE/MSE are standard practice in related work. That is useful context. However, my original concern is still only partly addressed. since the model is used mainly to produce similarity scores for BO and GRPO, I still think the paper would benefit from more direct evidence that these similarity scores are useful in the downstream pipeline.

---

> > > ### Author Response · Authors · 2026-04-03
> > >
> > > Thank you for your careful and constructive review! Incorporating your suggestions has guided us to add more details and evidence in the paper. Please find our detailed responses below.
> > >
> > > ### **1. Similarity Score**
> > >
> > > We understand your concerns, which are also relevant to the actual situation of research in the EDA field. In leaf cell design, since all data that can be collected is simply the layout itself, not the script, similarity scoring is actually a crucial and necessary step in helping GenLeaf establish a layout-script mapping. To further enhance our paper, we have added more direct evidence to demonstrate the effectiveness and practicality of similarity scoring, including **ablation experiments on the performance of BO-searched scripts** (the first table) under different representation model configurations and **the convergence process of BO** (the second table).
> > >
> > > | Configuration                       | Similarity Score to Golden | #Track | WL    | #Via  |
> > > | ----------------------------------- | -------------------------- | ------ | ----- | ----- |
> > > | Supervised Representation (Ours)    | 0.9637                     | 7.42   | 44.97 | 20.71 |
> > > | Representation w/o Similarity Score | 0.5302                     | 9.14   | 45.82 | 20.64 |
> > > | Unsupervised Representation         | 0.4971                     | 13.28  | 71.30 | 23.50 |
> > >
> > > As shown in the table above, supervision based on similarity scores is crucial; searching solely by physical metrics yields significantly worse results. The physical design of leaf cells requires considering routing during the placement phase, which necessitates expert experience. Therefore, blindly optimizing the three physical metrics may require more exploration to find a better solution and easily lead to local optima. However, by using the similarity score to provide PnR topological guidance, the representation model can converge to the golden layout much faster, thus effectively guiding the BO and GRPO processes.
> > >
> > > ### **2. BO Convergence to Golden Results**
> > >
> > > We utilize a **manually designed layout by experts as the golden layout** and employ a similarity-based BO for script search. As shown in Section C.1, the scripts searched by BO show an average difference of only 3.63% from the golden results. As shown in the table below, the scripts generated by the BO approach the golden layout with increasing accuracy in each iteration, ultimately converging to a high similarity. Furthermore, a more detailed BO convergence process is available in *BO_convergence.md* in the article's anonymous repository.
> > >
> > > | BO Iteration | Current Similarity | Best Similarity | Diff from Golden (%) |
> > > | ------------ | ------------------ | --------------- | -------------------- |
> > > | 50           | 0.6738             | 0.7587          | 32.62                |
> > > | 100          | 0.8232             | 0.8399          | 17.68                |
> > > | 150          | 0.8879             | 0.9037          | 11.21                |
> > > | 200          | 0.8976             | 0.9421          | 10.24                |
> > > | 250          | 0.9331             | 0.9637          | 6.69                 |
> > > | 300          | 0.9527             | 0.9637          | 4.73                 |
> > >
> > > ### **3. Dataset Distribution**
> > >
> > > In the EDA field, the data involved is usually confidential, making it extremely scarce and difficult to collect. Therefore, most EDA-related works can only use a limited number (often less than 14) of test cases. For leaf cells, there are no open-source datasets. Therefore, we collaborated with leading enterprises to obtain these valuable data from industry.
> > >
> > > To demonstrate the generalization of GenLeaf, we carefully selected test cases from the entire dataset. The so-called “unseen” simply means that GenLeaf did not encounter **similar design types** during training. In our work, GenLeaf is not confined to a specific design type, but handles both types of designs encountered and not encountered in SFT and GRPO well. To further prove the generalization, we additionally collect several new layouts for supplementary experiments. As shown in the following table, GenLeaf demonstrates very stable and excellent performance across all cases with both seen and unseen design types. Same as Table 3 in the paper, the metrics of expert manual design are 1.
> > >
> > > | Cases                               | #Track Ratio | WL Ratio | #Via Ratio |
> > > | ----------------------------------- | ------------ | -------- | ---------- |
> > > | With Unseen Design Types (20 cases) | 0.531        | 0.963    | 1.043      |
> > > | With Seen Design Types (16 cases)   | 0.524        | 0.957    | 0.994      |
> > >
> > > ### **Conclusion**
> > >
> > > If there are any further suggestions that you think would help strengthen the paper and bring it closer to the acceptance bar, we would greatly appreciate hearing them. Once again, we greatly appreciate your constructive guidance. And we kindly invite you to reconsider the Score in light of the substantial improvements made.

---

### Decision · Program_Chairs · 2026-04-30

**Decision:**

Accept (regular)

**Comment:**

This is a use-inspired ML paper for the important real-world problem of expert-level leaf cell layout generation for designing integrated circuits. The paper develops an LLM-based approach by formulating the task as script generation guided by expert knowledge. The overall approach and workflow consists of three parts:
1) Performance-aware layout representation that captures both placement topology and routing information
2) Bayesian optimization to recover expert scripts for supervised fine-tuning
3) Specialized GRPO procedure to optimize candidate scripts using rewards derived from physical metrics and similarity to expert layouts

Experiments on real data demonstrates the efficacy of the approach compared to LLM-based baselines.

Reviewers' liked the paper in general but also raised some important questions including generalization claims based on narrow experiments, ablation analysis of different components, and evaluation metrics. Authors' have mostly addressed all these concerns in the rebuttal phase.

I recommend accepting the paper and encourage the authors' to incorporate all the review comments / discussion to improve the final paper and also to provide direct evidence that similarity scores BO and GRPO are useful for downstream components.